# Genetic and metabolomic architecture of variation in diet restriction-mediated lifespan extension in *Drosophila*

Kelly Jin[1], Kenneth A. Wilson[2,3], Jennifer N. Beck[2], Christopher S. Nelson[2], George W. Brownridge, III[2,4], Benjamin R. Harrison[1], Danijel Djukovic[5], Daniel Raftery[5], Rachel B. Brem[2,3,6], Shiqing Yu[7], Mathias Drton[8], Ali Shojaie[9], Pankaj Kapahi[2,3], Daniel Promislow[1,10]*

**1** Department of Pathology, University of Washington School of Medicine, Seattle, Washington, United States of America, **2** Buck Institute for Research on Aging, Novato, California, United States of America, **3** Davis School of Gerontology, University of Southern California, University Park, Los Angeles, California, United States of America, **4** Dominican University of California, San Rafael, California, United States of America, **5** Northwest Metabolomics Research Center, Department of Anesthesiology and Pain Medicine, University of Washington, Seattle, Washington, United States of America, **6** Department of Plant and Microbial Biology, University of California, Berkeley, Berkeley, California, United States of America, **7** Department of Statistics, University of Washington, Seattle, Washington, United States of America, **8** Department of Mathematics, Technical University of Munich, Munich, Germany, **9** Department of Biostatistics, University of Washington, Seattle, Washington, United States of America, **10** Department of Biology, University of Washington, Seattle, Washington, United States of America

* promislo@uw.edu

**Data Availability Statement:** All relevant data are within the manuscript and its Supporting Information files.

## Abstract

In most organisms, dietary restriction (DR) increases lifespan. However, several studies have found that genotypes within the same species vary widely in how they respond to DR. To explore the mechanisms underlying this variation, we exposed 178 inbred *Drosophila melanogaster* lines to a DR or *ad libitum* (AL) diet, and measured a panel of 105 metabolites under both diets. Twenty four out of 105 metabolites were associated with the magnitude of the lifespan response. These included proteinogenic amino acids and metabolites involved in α-ketoglutarate (α-KG)/glutamine metabolism. We confirm the role of α-KG/glutamine synthesis pathways in the DR response through genetic manipulations. We used covariance network analysis to investigate diet-dependent interactions between metabolites, identifying the essential amino acids threonine and arginine as "hub" metabolites in the DR response. Finally, we employ a novel metabolic and genetic bipartite network analysis to reveal multiple genes that influence DR lifespan response, some of which have not previously been implicated in DR regulation. One of these is *CCHa2R*, a gene that encodes a neuropeptide receptor that influences satiety response and insulin signaling. Across the lines, variation in an intronic single nucleotide variant of *CCHa2R* correlated with variation in levels of five metabolites, all of which in turn were correlated with DR lifespan response. Inhibition of adult *CCHa2R* expression extended DR lifespan of flies, confirming the role of *CCHa2R* in lifespan response. These results provide support for the power of combined genomic and metabolomic analysis to identify key pathways underlying variation in this complex quantitative trait.

**Funding:** KJ was funded by the NIH/NIA T32 Genetic Approaches to Aging Training Grant AG000057. KAW was funded by the NIH/NIA F31 grant AG052299. RBB was funded by R01 GM087432. AS was funded by NIGMS grant GM114029. PK was funded by grants from the American Federation of Aging Research and Hillblom foundations and NIH grants R56 AG038688 and AG045835. DP was funded by NIA grants R01 AG049494 and P30 AG013280. DP, MD, and AS were funded by NSF grant DMS1561814. The funders had no role in study design, data collection and analysis, decision to publish, or preparation of the manuscript.

**Competing interests:** The authors have declared that no competing interests exist.

## Author summary

Dietary restriction extends lifespan across most organisms in which it has been tested. However, several studies have now demonstrated that this effect can vary dramatically across different genotypes within a population. Within a population, dietary restriction might be beneficial for some, yet detrimental for others. Here, we measure the metabolome of 178 genetically characterized fly strains on fully fed and restricted diets. The fly strains vary widely in their lifespan response to dietary restriction. We then use information about each strain's genome and metabolome (a measure of small molecules circulating in flies) to pinpoint cellular pathways that govern this variation in response. We identify a novel pathway involving the gene *CCHa2R*, which encodes a neuropeptide receptor that has not previously been implicated in dietary restriction or age-related signaling pathways. This study demonstrates the power of leveraging systems biology and network biology methods to understand how and why different individuals vary in their response to health and lifespan-extending interventions.

## Introduction

No two individuals age in exactly the same way [1, 2]. Variation in aging, as with all complex traits, is determined by contributions from variation in genes, the environment, and the interaction between the two. Similarly, an individual's response to interventions that slow aging is likely to be equally as complex.

Among these interventions, dietary restriction (DR) has been shown to extend lifespan in almost all model organisms in which it has been tested [3]. However, despite this high level of conservation across species [4], several studies suggest that the DR response might be far from universal. For example, Harper *et al.* [5] found no effect of DR on mean longevity in grand-offspring of wild-caught mice. Additionally, results from studies in three different species reveal extensive within-species variation in the DR response [6–10]. Liao *et al.* [6] showed that fewer than half of 41 recombinant inbred mouse strains on DR showed a lifespan increase. Several years later, a study of 166 single-gene deletion yeast strains found variation in the DR response, ranging from a 79% reduction to a 103% increase in lifespan [7]. In 2017, Stanley *et al.* [9] measured lifespan for 80 recombinant inbred *Drosophila melanogaster* lines on control and DR diets and found highly significant diet-by-genotype interaction for median life span. Most recently, Wilson *et al.* [10] showed that of 161 naturally-derived inbred lines of *D. melanogaster*, 71% showed a DR-mediated lifespan extension, while the rest showed either a non-significant lifespan response or reduced lifespan under DR ([10], Fig 1A and S1A Fig). In light of this considerable variation, to fully understand the factors that determine the response to DR, we need to understand not only the mechanisms underlying DR itself, but also the mechanisms that influence *variation* in the DR response within a population.

To better understand why some individuals respond to DR while others do not, we employ a systems biology approach, focusing on the influence of two distinct biological domains—the genome and the metabolome. Given that DR dramatically changes the metabolic state of an organism, metabolomic profiling has become an invaluable tool for investigating the mechanisms underlying DR and aging [11–13]. The metabolome, which includes all low-molecular weight molecules within a biological system, plays a crucial role in the biology of aging [14]. Examples include the role of specific amino acids like tryptophan or methionine in DR [15, 16], of polyamines in age-related decline in circadian periodicity [17], and of α-ketoglutarate

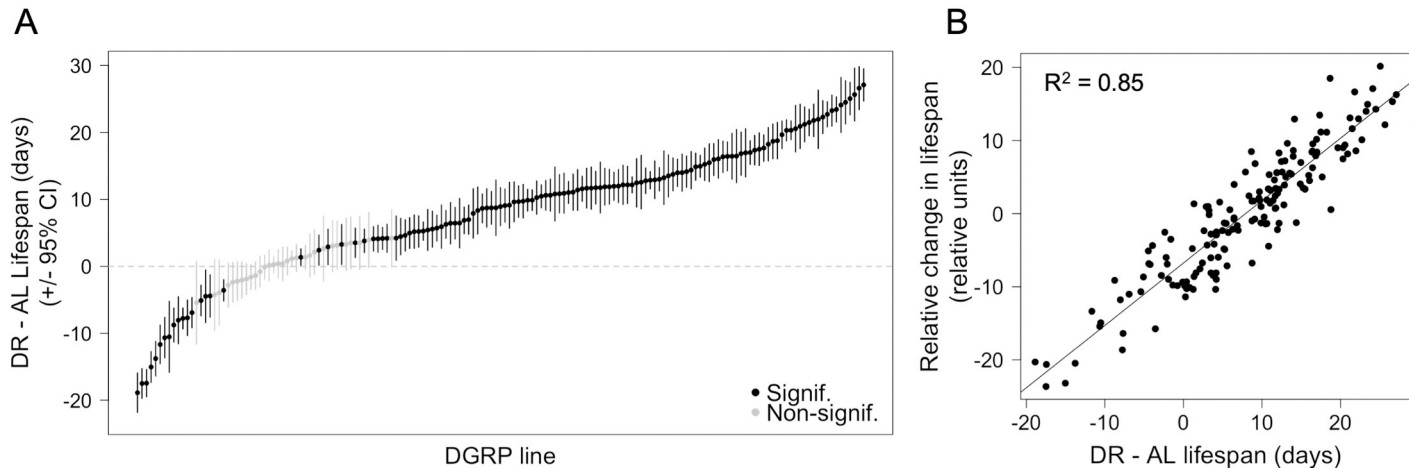

**Fig 1. Variation in DR-mediated lifespan extension across the DGRP.** (A) Variation in DR–AL lifespan measured across 161 DGRP lines plotted in ascending order. Each point represents a fly line. Statistical significance was determined using 5% FDR adjusted *P* value from Student's t-tests. Error bars represent 95% confidence interval from t-test. (B) Relationship between change in lifespan and relative change in lifespan (rLS). The two lifespan traits are significantly correlated.

(α-KG) and its derivatives in TOR signaling and epigenetic regulation [18, 19]. High-through-put, high-resolution metabolomic profiling methods allow us to capture a snapshot of the circulating products and intermediates of cellular metabolism within a tissue or organism. This ability has proven invaluable for understanding the mechanisms underlying complex traits for many reasons. First, although feedback mechanisms likely exist between all domains within a biological system, the metabolome is generally thought to be downstream of transcriptional and translational regulation. Second, the metabolome is highly influenced by, and therefore integrates information from, both the genome and environment [11, 20]. As a result, the metabolome may explain a greater proportion of phenotypic variation within a population than genomic, transcriptomic, or proteomic profiling alone. This is critical, as many genome-wide studies of complex traits explain only a fraction of the phenotypic variation [21]. Third, metabolites are involved in all biological processes within the cell, and as such, metabolomic studies can help bridge the gap between genotype and phenotype [20, 22] [23–26]. As researchers learn more about the intricacies of the various cellular mechanisms that govern aging, systems biology approaches such as these are essential for revealing mechanisms that underlie these complex phenotypes.

Our group previously showed that DR slows age-related changes in the metabolome and dramatically alters metabolic network structure in a single, wildtype strain of *Drosophila* [11]. However, this work did not consider the extent to which the response to DR varies among genotypes. Here, we explore the metabolic signatures from different fly strains that vary widely in their response to DR by profiling the metabolome in 178 inbred fly lines from the *Drosophila* Genetic Reference Panel (DGRP), a set of inbred, fully sequenced strains of *Drosophila* derived from a wild population [27, 28]. The DGRP has been successfully used to profile and genetically map complex quantitative traits including longevity [29], oxidative stress resistance [30], microbiota composition [31], and many others [32].

Here, building on a prior analysis of lifespan measurements collected by Wilson *et al.* [10], we extend this with metabolomic profiling of flies from the same experiment. Taken together, this is the largest *Drosophila* study we know of to leverage both the metabolome and genome of a large number of genotypes of the same species to study lifespan response to DR. Specifically, we show that i) DR dramatically remodels the metabolome in consistent ways across 178

different genotypes; ii) several individual metabolites correlate with lifespan response, including amino acids and metabolites involved in α-KG/glutamine metabolism; iii) differential network analysis reveals that metabolite network structure varies with diet and reveals 'hub' metabolites that gain or lose multiple network connections in a diet-specific manner; iv) network analysis of genome-wide association tests reveals candidate gene-to-metabolite-to-phenotype pathways that might underlie the DR response; and v) down-regulation of three of the genes identified from this analysis alters lifespan response, including a neuronal effect of *CCHa2R*, a gene that encodes a neuropeptide receptor thought to be involved in nutrient sensing and satiety response.

## Results

### Diet specific changes in the metabolome

Targeted metabolomic profiling was performed on 178 DGRP lines for each diet condition. The flies used for metabolomics were collected from the same cohort of flies used for lifespan measurements. The lifespan measurements across the DGRP on AL and DR were first presented and discussed in Wilson *et al.* [10], while the metabolomic data are novel to this study.

After quality filtering (see Methods), 105 metabolites were included in downstream statistical analysis. With this dataset, our goal was to identify biologically meaningful relationships that might exist between diet, lifespan response, the genome, and the metabolome.

To begin, we summarize the effect of DR on the metabolome across the DGRP lines using principal component analysis (PCA). The first principle component (PC) explained 69% of the variation across the entire metabolome, and cleanly separated samples by diet, revealing that DR has an extremely strong effect on the fly metabolome (S2A Fig).

To determine which metabolites were affected by diet, difference in mean abundance (DR–AL) was calculated for each metabolite across all the lines. Since this calculation was performed on log-normalized and scaled data (see Methods), we interpret the resulting difference as a *relative* change in abundance (i.e., log(DR)–log(AL) = log(DR/AL)). As determined from paired t-tests, almost all measured metabolites showed a highly significant response to diet (S2B Fig, Table A in S1 Table). These results are consistent with previous studies demonstrating a substantial remodeling of the fly metabolome under DR [11].

### The effect of DR on lifespan in the DGRP

There is considerable variation in lifespan response as a result of DR across the DGRP (S1A Fig, [10]). As expected, we also observed a wide range of variation across the DGRP in the difference in lifespan between diets (DR–AL) (Fig 1A). Change in mean lifespan varied among lines from 18 days shorter to 27 days longer in response to DR. Of the 161 DGRP lines that have lifespan measurements, 114 lines (70.8%) lived longer, 30 lines (18.6%) did not show any significant change in lifespan, and 17 (10.5% lines) lived shorter as a result of DR (Fig 1A, [10]).

The primary goal of this study is to investigate the lifespan response to DR across the DGRP. The difference between DR and AL lifespan is one measure of this response. However, this difference value is correlated with AL lifespan (S1C Fig), which means that any significant associations that we find with the difference in lifespans might actually be due to a relationship with AL lifespan. To remove potentially confounding effects of AL lifespan on the DR response, we derive another lifespan response trait that we call the relative change in lifespan (rLS; see Methods). rLS is highly correlated with the difference in lifespans (Fig 1B), but not with AL lifespan (S1D Fig). For the remainder of this study, we will use rLS as our primary lifespan response phenotype.

## Metabolites are significantly correlated with lifespan response

To determine which metabolites are correlated with lifespan traits (mean lifespan and rLS), we used a simple linear regression to model lifespan phenotypes as a function of each metabolite measured under either diet:

$$\text{Lifespan trait} = \alpha + \beta \cdot \text{metabolite}_{AL} + \varepsilon \tag{1}$$

or

$$\text{Lifespan trait} = \alpha + \beta \cdot \text{metabolite}_{DR} + \varepsilon \tag{2}$$

Looking first for metabolites associated with mean lifespan, we found that 10 metabolites measured under DR were correlated with mean DR lifespan (with a false discovery rate (FDR) = 0.01) (Fig 2A, Table B in S1 Table). Notably, no metabolites measured under AL were found to be correlated with mean AL lifespan (Fig 2A; Table C in S1 Table), though 8 metabolites measured under AL were correlated with DR lifespan (Table C in S1 Table).

We next identified metabolites that were correlated with rLS. Out of a total of 105 measured metabolites, 24 were associated with rLS after FDR correction (Fig 2B, Table B and C in S1 Table). Thirteen metabolites are associated with rLS under both conditions (top right quadrant in Fig 2B), while 10 were only significant when measured under AL (bottom right quadrant in Fig 2B), and only threonine was significant under DR (top left quadrant in Fig 2B).

Next, we asked if the *change* in metabolite abundance was predictive of lifespan response. To do this, we modeled rLS as a function of change in metabolite level ($\Delta mz_i = mz_{DR,i} - mz_{AL,i}$ for metabolite $i$). Of 105 metabolites, the change in abundance of 4 metabolites—2-phosphoenolpyrivic acid (PEP), 2-phosphoglyceric acid, threonine, and arginine—was associated with rLS (Fig 2C, Table D in S1 Table).

## Diet-dependent changes in metabolite networks

Having identified metabolites associated with lifespan and its response to DR across genotypes, we wanted to investigate the relationship between these metabolites and the broader metabolome measured here, and more specifically, to determine how those relationships change between diets. To capture this, we measured covariance among all metabolites and performed differential network analysis to investigate interactions between metabolites across diets (Fig 3A). We grouped interactions into three categories: 1) metabolite-metabolite interactions that are significantly greater (as measured by correlation coefficient $\rho$; see Methods) under AL than DR (red edges in Fig 3A; Fig 3B); 2) interactions that are significantly greater under DR than AL (blue edges in Fig 3A; Fig 3C); and 3) interactions that are strongly significant under both conditions (yellow edges in Fig 3A; Fig 3D). We focus solely on positive correlations because there were no significant negative correlations between metabolites in the network.

Some metabolites gain or lose many interactions under one diet in comparison with the other. Of these, the most striking example is arginine, which gains 14 edges under AL (Fig 3E and right asterisk in Fig 3A). Such metabolites may represent 'hubs' in the metabolome network. Hubs are also found under DR conditions, with threonine gaining seven edges (Fig 3F and left asterisk in Fig 3A). Three features of this metabolome network stand out to us. First, the two hub metabolites threonine and arginine are among the top hits for $\Delta mz$s associated with rLS (Fig 2C). Second, many of the edges that these hub metabolites gain or lose across diets are shared with other amino acids (Fig 3A). Last, this network includes many of the essential amino acids, most of which are all correlated to one another under both AL and DR (boxed metabolites in Fig 3A). The essential amino acids in *Drosophila* include arginine and the nine essential amino acids in humans [33].

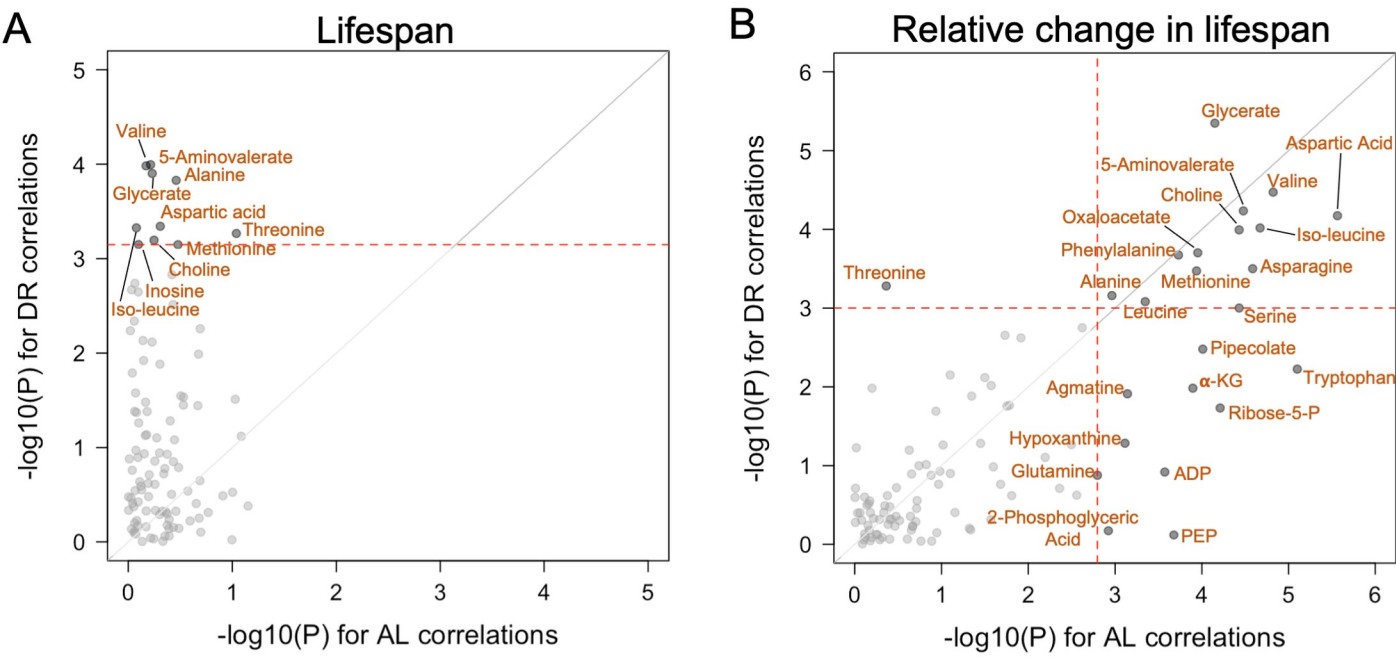

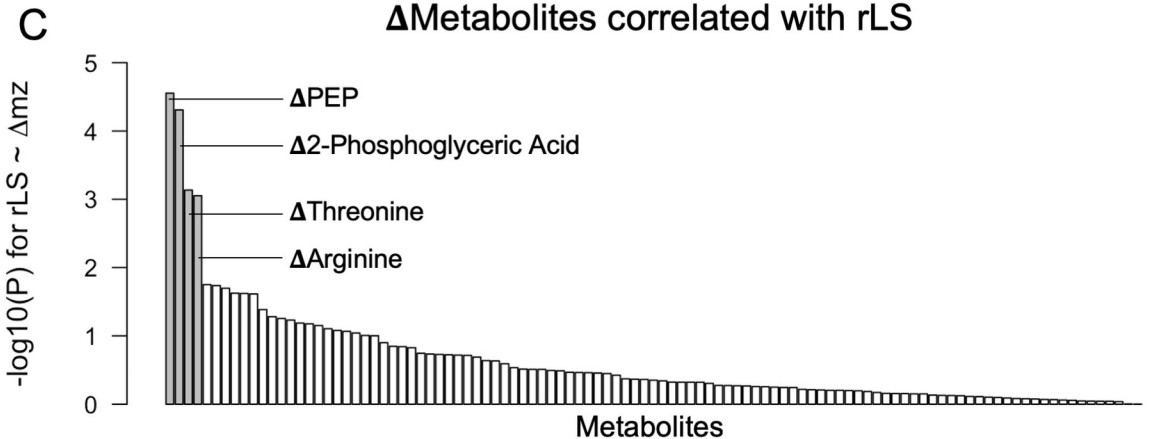

**Fig 2. Metabolites significantly correlated with lifespan response.** (A,B) Results of univariate analysis modeling lifespan phenotypes as functions of individual metabolites measured under either AL or DR were performed and $-\log_{10}(P$ values) were plotted. Each point represents a single metabolite and the significance of its association with (A) mean lifespan and (B) rLS. Red dotted lines represent FDR cutoff at $\alpha = 0.01$. (C) Rank of $-\log_{10}(P$ values) from linear regression modeling rLS as a function of change in metabolite abundance. Four labeled metabolites passed FDR cutoff of $\alpha = 0.05$.

### Manipulation of α-KG pathways alters lifespan response to DR

Our analysis of metabolites associated with the lifespan response to DR identified both α-KG and glutamine (Fig 2), which form a sub-network within the overall metabolome network (Fig 3A and 3D). Given our finding, and previous studies implicating α-KG in DR and TOR signaling [18], we used RNA interference (RNAi) to knock down genes in the α-KG/glutamine pathway to explore its possible role in the effect of DR on lifespan. Specifically, we manipulated the expression of glutamate dehydrogenase (GDH), which catalyzes the reversible conversion between αKG and glutamate, and glutamine synthetase 1/2 (GS1, GS2), which catalyze the conversion of glutamate to glutamine (S3A Fig). For these experiments we used the inducible

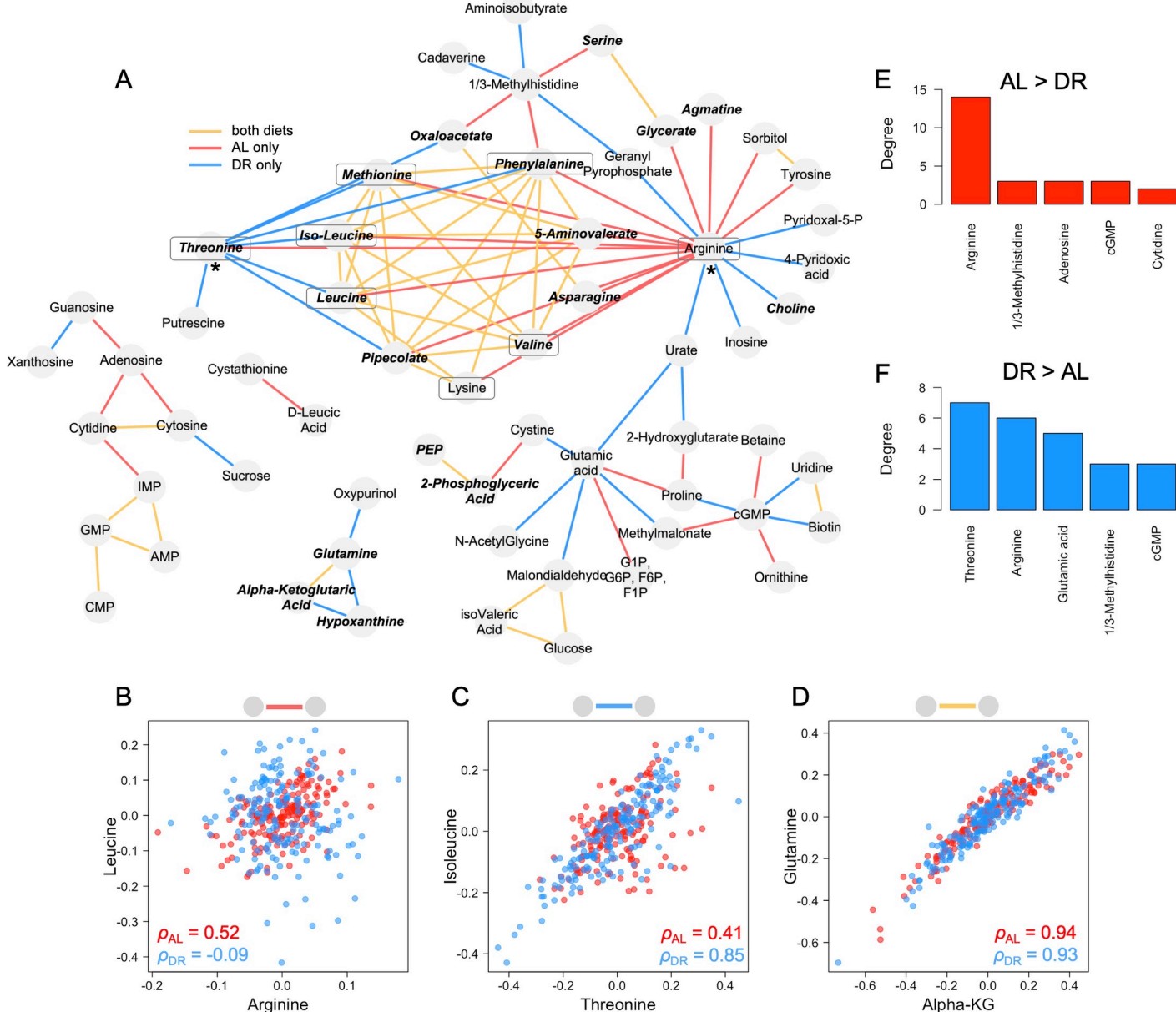

**Fig 3. Diet-dependent correlation network analysis.** (A) Nodes represent metabolites and edges represent correlation between two metabolites. Edge color denotes correlations that significantly become more positive with AL (red), DR (blue), or have correlation coefficients that exceed abs(0.8) in both diets (yellow). For red and blue edges, only correlation coefficient differences of greater than abs(0.4) are shown. Bolded and italicized metabolites were also found to be significantly associated with rLS. Boxed metabolites are essential amino acids. Asterisks indicate suspected "hub" metabolites. (B-D) Examples of each type of covariance relationship (edge color) are shown. (E-F) Degrees (number of edges) of top 5 most connected nodes under each diet.

GAL4-GeneSwitch (GAL4-gs) drivers to knock down each gene in different parts of the fly upon induction with the compound RU486 [34].

We exposed inducible RNAi fly strains to AL and DR diets and recorded lifespan. To test for a change in lifespan response as a result of RNAi, we tested for a significant RNAi-x-diet interaction term in a Cox Proportional Hazards model (see Methods). Knockdown of *gs2* in the whole fly most strongly affected the lifespan response as compared to knockdown of *gs1* and *gdh*, with knockdown flies living longer on the AL diet (S3B–S3D Fig). A greater extension

of AL lifespan was seen with neuron-specific knockdown of GS2, but less so in fat body or gut-specific RNAi (S3E–S3G Fig). This suggests that GS2 is part of a diet-specific lifespan pathway active in neurons.

## Gene-metabolite-phenotype network analysis for lifespan response

While we found that fully 23% of measured metabolites were associated with rLS, we performed a gene-level genome-wide association study (GWAS) on the rLS trait and only one gene, an uncharacterized coding gene CG6231, was associated with rLS. In light of this, we decided to leverage the metabolite-lifespan associations identified here, creating a bipartite network of genes and metabolites that would allow us to look for genes that might impact lifespan indirectly through their impact on the metabolome. To do this, we first performed multiple individual gene-level GWAS using metabolite levels as quantitative traits. Specifically, we selected metabolites that were associated with rLS. This included 23 AL metabolites and 14 DR metabolites (Fig 2B). The gene-level GWAS resulted in significance scores for each gene-metabolite pair. Briefly, gene scores were assigned by taking the minimum *P* value from all variants associated with a gene after adjustment via at least 10,000 rounds of permutation testing. A detailed explanation of this calculation is included in the Methods. Significant gene-metabolite relationships were used to build a network connecting genes, metabolites, and the lifespan response phenotype. The resulting multi-omic network for AL metabolites is visualized in Fig 4 and the network for DR metabolites in S4 Fig. All genes that appear in the DR network also appear in the AL network, so we display the AL network in the main figures. The purpose of this network is to diagram molecular paths that might regulate lifespan extension, starting from the gene level (teal nodes) to the metabolite level (yellow nodes), and that ultimately influence lifespan response to diet as quantified by rLS (center grey node). The degree, or number of edges, of each gene node is represented by its size in the figure. A complete list of significant AL and DR network edges and scores is provided in Table E in S1 Table.

Across both the AL and DR network, the gene with the most significant gene-level score was *Vps15* which is associated with AL levels of 2-phosphoglyceric acid and PEP (Table E in S1 Table). *Vps15* encodes a serine/threonine protein kinase that is part of the PI3-Kinase (PI3K) complex [35]. We can also examine gene node degree as an indicator of the gene's role in the signaling network. It is important to note that the degree of a gene node can be interpreted to mean that a gene affects all metabolites its connected to independently, or that a gene more directly affects the level of one metabolite whose level in turn is correlated with other metabolites. In the latter case, some edges may represent indirect gene-metabolite relationships. For example, *CCHa2R*, which has a degree of 5 in the AL network (Fig 4), could independently associate with five metabolites. However, it may instead affect the abundance of one metabolite more directly than the others. In both the AL and DR networks, the gene with the largest degree value is *CCHa2R* (Fig 4, S4 Fig). *CCHa2R* encodes the protein CCHamide-2 receptor (CCHa2-R), a neuropeptide receptor thought to be expressed in the brain and gut of fruit flies, and is proposed to be involved in appetite regulation and insulin signaling [36, 37]. Thus, our network analysis identified genes that might be involved in regulating lifespan response to DR, including *Ddc*, *SelT*, *jeb*, *Nuak1*, *z*, and several uncharacterized coding genes (Fig 4, S4 Fig, Table E in S1 Table).

## *CCHa2R* and *SelT* are involved in the lifespan response to DR

To test the ability of our multi-omic network analysis to identify genes involved in the DR response, we used inducible RNAi to measure the effects of gene knockdown on AL and DR lifespans. We focused on the genes *CCHa2R* and *selT*. The single nucleotide polymorphism

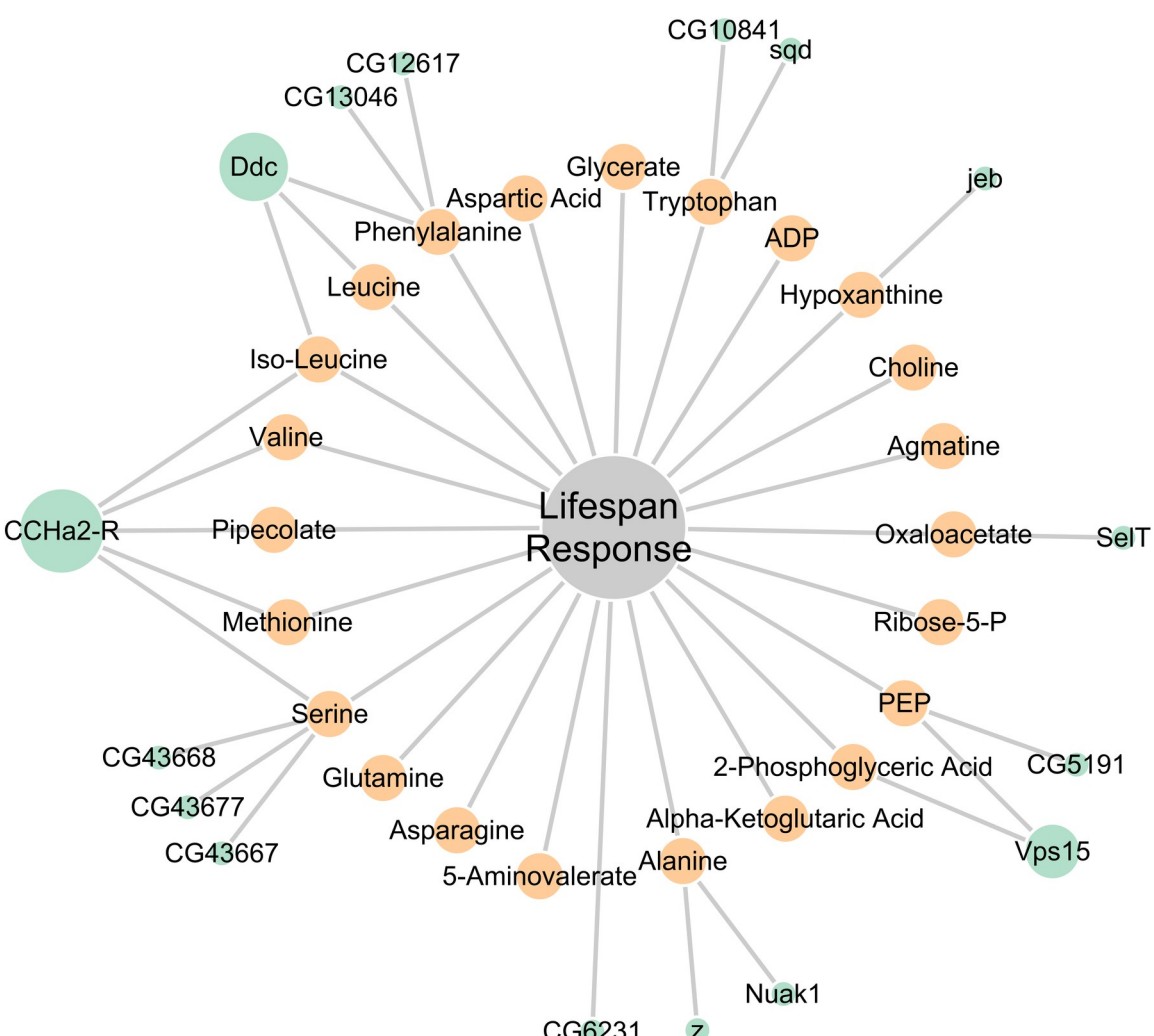

**Fig 4. Multi-omic network for lifespan response.** Gene-metabolite-phenotype network was constructed from linear modeling and GWAS results from AL metabolites that were correlated with lifespan response as measured by rLS. Gene nodes are colored in teal, metabolite nodes are colored in yellow. Gene node size is directly proportional to node degree, while metabolite node size is held constant. An edge exists between a metabolite and lifespan response if the metabolite was significantly correlated with rLS at FDR cutoff of α = 0.01. An edge exists between a gene and metabolite and/or lifespan response if it the gene had a score of ≤1E-4.5.

(SNP) 2R_1939249_SNP identified from metabolite GWAS is a C/T variant in an intron of *CCHa2R*. This SNP was associated with differential abundance of five rLS-associated metabolites under AL (Fig 5A). As an example, the relationship between iso-leucine, rLS, and *CCHa2R* SNP are shown in Fig 5B–5D. Knocking down *CCHa2R* using a whole-body driver resulted in an increased mean lifespan under DR, but a slightly decreased mean lifespan under AL, although this effect was not statistically significant under a Cox Proportional Hazards model framework (Fig 5E). Given this trend, we then used a neuronal specific driver to knock down *CCHa2R*, which showed a much greater effect of increasing lifespan under DR, but not AL (Fig 5F).

Manipulation of *SelT* also resulted in changes to diet-specific lifespans (S5 Fig). *SelT* encodes selenoprotein T. Selenoproteins are a family of thioredoxin-disulfide reductases that plays a role in defending against cellular oxidative damage by controlling the redox balance of the cell [38]. We found that whole-body *SelT* knockdown increased lifespan on AL while

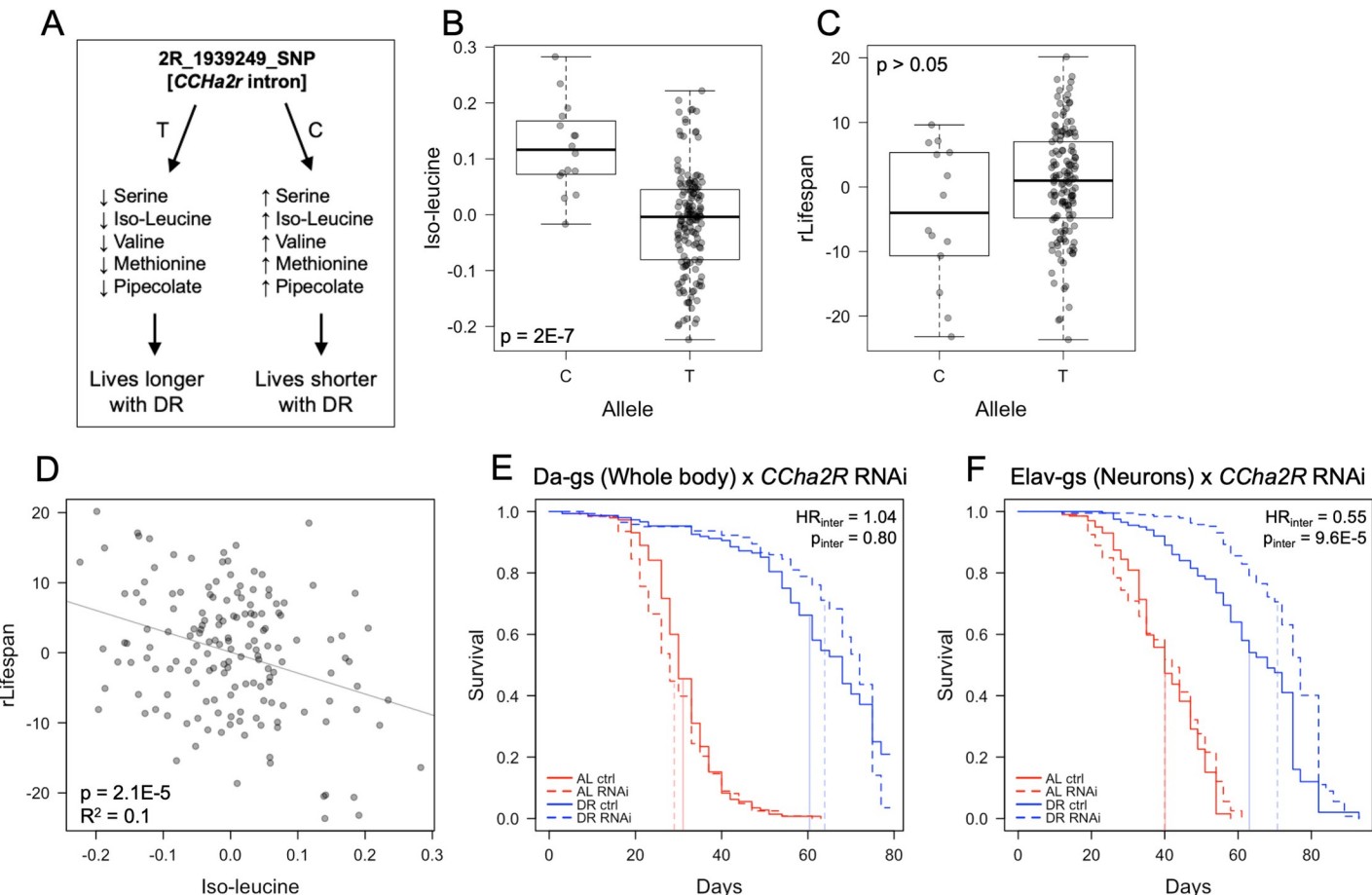

**Fig 5. *CCHa2r* is associated with change in metabolite levels and modulates lifespan response to DR.** (A) Diagram of one of the candidate gene pathways identified from metabolite-gene-network analysis, *CCHa2r*, and its relationship with iso-leucine and lifespan response. (B-D) Iso-leucine and its relationship with *CCHa2r* SNP 2R_1939249_SNP and residual lifespan. (E-F) Survival of RNAi (+RU486) versus control (-RU486) flies of *CCHa2r* RNAi in whole-body (D; da-gal4-gs driver) and neurons (E; elav-gal4-gs driver). Vertical lines represent mean lifespan. All lifespan experiments were conducted with 150–200 flies per condition. *P* values from B and C are from plink linear GWAS model. *P* value from D is from a linear regression as summarized in Fig 2. Statistical model in E and F is a Cox Proportional Hazards model fitting survival as a function of diet, RNAi, and the interaction between diet and RNAi. Hazard ratios (HR) and P values are specific to the interaction term.

having no effect on lifespan under DR (S5 Fig). Taken together, these results support our network analysis as a map of gene-to-metabolite-to-phenotype pathways that underlie natural variation in the DR response and also highlight the fact that DR response can be influenced through a change in AL lifespan, DR lifespan, or both.

## Discussion

Diet restriction is regarded as the most robust form of lifespan extension known, and has been consistently demonstrated to increase longevity in almost all model organisms in which it has been studied [3]. However, in almost all of these studies, researchers have used a single strain chosen from a handful of common lab-adapted strains, limiting our ability to gain insight into variation in response to DR in a diverse population. Here, we have leveraged the power of the genetic variation found within the DGRP, together with systems biology methods, to measure and explain genetic variation in the DR response. This work brings three key approaches to bear on the study of DR response, including natural genetic variation, metabolome profiling, and network modeling. By combining all three of these, an approach never used before in the

context of DR, we are able to generate novel genetic and biochemical hypotheses about DR. We then validate these findings experimentally using inducible RNAi.

This work adds to a growing body of evidence for genetic variation in the DR response, including similar studies conducted in mice [6], yeast [7], and a different fly genetic reference panel derived from recombinant inbred strains [9]. While these previous studies underscore the importance of genetic variation in shaping the DR response, our work focuses in particular not on single-gene knock-out strains or genotypes derived from recombinant inbred strains, but rather on the natural genetic variation derived from a single wild population [28]. The translational path from *Drosophila* to humans is a long one, but these results do underscore the possibility that the effect of interventions designed to decrease or delay the onset of age-related decline in human populations could have diverse outcomes from one person to the next, depending on individual genetic makeup and environment history [6–9]. The lifespan results we present in this study were also analyzed by Wilson *et al* [10]. We note that Wilson *et al.* found multiple candidate variants that affect diet-dependent longevity, while we identify one that met our significance cutoff (Fig 4 and S4 Fig). This is not surprising, given that these two studies fit two different response variables to different covariates. Wilson *et al.* used a linear model predicting mean lifespan with an interaction term between genotype and diet for their GWAS, while we use relative lifespan as the outcome for our GWAS.

While it is relatively straightforward to measure genetic variation for complex traits, identifying the individual genes that contribute to this variation has proven to be far more challenging [21]. To fill this gap, researchers have turned to the metabolome. This has helped not only to define genetic variation, but also to suggest mechanisms that underlie this variation [23–26]. Previous studies have found that DR leads to dramatic shifts in metabolism in diverse organisms [39] and that it attenuates metabolic signatures of aging [11, 40, 41]. Metabolomic profiling has become a popular tool for investigating mechanisms underlying DR within single genotypes across a wide array of species, including worms [41, 42], flies [11, 43], mice [40, 44], non-human primates [45], and humans [46, 47]. Many of these studies pointed to changes in fatty acid metabolism as a result of DR [40, 44, 47]. This is consistent with the finding from Liao *et al.* that across 41 mouse strains, the ones with the least reduction in fat under DR were more likely to show lifespan extension on DR [48]. Our study, which is the first that we know of to use metabolome profiling to investigate lifespan response to DR across genotypes, identified 25 metabolites that were correlated with lifespan response either through their baseline abundance or change in abundance across the two diets (Fig 2B and 2C). We did not profile enough lipid metabolism-related metabolites to be able to test whether or not fat maintenance might be an influential component underlying the DR response. However, one of the genes our bipartite network analysis identified was *jeb* (jelly belly), which encodes a low-density lipoprotein proposed to be involved in neuronal PI3K signaling during nutrient restriction in flies [49]. Jelly belly may regulate lifespan response through a mechanism involving the metabolite hypoxanthine (Fig 4). In addition, many of the 25 metabolites associated with lifespan response in our study were related to amino acid metabolism, which we have previously shown to be a significant pathway modified under DR in flies [11].

As we note above, the metabolome can act as a statistical link between genotype and phenotype [25, 26]. Here we advance this approach in two important respects. First, we use a novel differential network analysis to contrast AL and DR metabolite networks in a manner that allows straightforward comparison between the two diet conditions (Fig 3), providing the critical and necessary context needed to interpret the interactions that we see under DR. This network captures a snapshot of metabolic relationships that might explain how flies translate nutritional environment to lifespan. For example, others have shown that restricting essential amino acids, methionine in particular, is necessary and sufficient to extend lifespan in flies

[50]. Our differential network links methionine and other essential amino acids to metabolites that correlate with lifespan response, diagramming metabolic paths that might explain the relationship between methionine, other amino acids, and the DR response (Figs 3 and 4 and S4 Fig). Second, while previous studies have typically focused on single molecular domains such as genetics or metabolomics [51–54], here we create a *bipartite network* that links specific genes to metabolites, and specific metabolites to the DR response, allowing us to identify new candidate genes and pathways that we begin to explore here. We use molecular genetic methods to confirm that indeed, many of the genes we identified influence lifespan response under dietary modification, including *CCHa2R* and *SelT*. We also identify other genes that have been identified by previous independent studies of *Drosophila* longevity. One of these genes, *Ddc*, encodes dopa decarboxylase, which has been suggested to influence natural variation in *Drosophila* longevity via a neuronal mechanism [55], adding further credence to the validity of our network approach. Other genes that will be of particular interest to pursue in future studies include *jeb*, as previously discussed, and *Vps15*, a PI3-Kinase regulator involved in autophagy.

In addition, our RNAi results suggest a role for neuronal perception of nutrient availability in lifespan response to DR. In *Drosophila*, neuronal signaling regulates physiological response to environmental nutrients, including gustatory and metabolic perception of amino acids and sugars [56–58], receipt of signals about nutrient availability from peripheral organs such as the gut [59], and feeding restraint [60]. We present evidence for a neuron-specific mechanism involving glutamine/α-KG signaling that influences lifespan under AL. GS2 RNAi, but not GS1, extended the lifespan of flies on AL (S3 Fig). In *Drosophila*, GS1 localizes primarily to the mitochondria while GS2 localizes to the cytosol [61]. In light of this, we postulate that cytosolic glutamine and perhaps glutamatergic signaling in neurons play a role in the DR response. This idea is supported by recent findings identifying a set of glutamatergic interneurons that signal *Drosophila* larvae to overcome amino acid limitation and pupate [62]. Furthermore, support for the suggestion that regulation of α-KG levels partially explains the lifespan response we observed is provided by work in *Caenorhabditis elegans* showing diet-dependent regulation of mTOR signaling by α-KG [18]. This evidence suggests the existence of interactions between glutamine/α-KG signaling and mTOR that ultimately influence lifespan response to DR.

We also found that knocking down expression of neuronal *CCHa2R* resulted in longer lifespan under DR (Fig 5). To our knowledge, this is the first study to implicate *CCHa2R* in an aging/DR-related signaling mechanism. We hypothesize that *CCHa2R* is associated with lifespan due to its influence on metabolism, an idea supported by previous studies in *Drosophila*. CCHa2-R is a G-protein coupled receptor that exclusively binds the neuropeptide CCHa2 and is thought to be involved in nutrient sensing and satiety response [36]. CCHa2 is mainly expressed in the fat body and at low levels in the gut and central nervous system (CNS), while its receptor CCHa2-R is highly enriched in the CNS, particularly in *Drosophila* insulin-like peptide (Dilp) producing cells that control the secretion of neuropeptide F and SIFamide in the brain [36]. In fly larvae, CCHa2 signaling appears to mediate the secretion of Dilp2 and Dilp5 [36] and signaling between peripheral organs and the brain [37]. Given the important role of Dilps both in nutrient signaling and aging [63], this novel connection between CCHa2-R and the effect of DR on aging is not surprising. In addition, the closest human homolog of CCHa2-R is bombesin receptor subtype 3 [64]. Mammalian bombesin-like peptides are widely distributed in the central nervous system and gastrointestinal tract, mirroring the distribution of CCHa2 peptide in *Drosophila* [36], and are thought to mediate signaling between the gut and brain, regulating processes such as smooth-muscle contraction, metabolism, and behavior [65]. The striking similarities between *Drosophila* CCHa2 signaling and mammalian bombesin signaling suggest that the processes are conserved to some extent.

Studying the genetic basis of natural trait variation complements existing mutational lab studies, offering critical additional insight into the biology of DR. Like other GWAS of lifespan in the DGRP [29, 66], we sought out to identify natural genetic variation affecting phenotypic variation. Interestingly, the genes and pathways identified by our work along with Durham *et al.* [66] and Ivanov *et al.* [29] largely differ in comparison to the canonical lifespan-associated genes and pathways identified by mutational studies. Although the reasons for this are unclear, these differences further emphasize the importance of systems approaches such as ours for developing a more complete picture of how genetics and environment impact healthy aging.

## Limitations of study

Readers should keep two caveats in mind in evaluating the results presented here. First, DGRP metabolite profiles were measured from whole fly bodies. However, different tissues of the fly, including the head, thorax, and abdomen, show different metabolome profiles under AL and DR [11]. Given the large number of lines measured here, tissue-specific profiling across the DGRP was outside the scope of the present study. In light of our understanding that neuronal perception of nutrients plays an important role in aging, future studies focused on head or brain specific metabolomics would be of great interest (e.g.[67]).

Second, flies were diet restricted by decreasing the percentage of yeast extract in their food. This diet has been used previously to demonstrate the effects of DR in flies [68–70]. There are many different methods of implementing DR that have been published in the *Drosophila* community, [50, 71, 72], and no single diet is considered standard, raising challenges in comparing results across studies. Furthermore, some have argued that different fly strains might have different optimal DR food levels, and that a decrease in fecundity must coincide with increased lifespan in order for the response to truly be considered DR [73]. Future work in this area should include genetic variation for survival *and* fecundity, should explore different types of DR, and finally should consider whether genotypes differ in how they respond to DR because they vary in their optimal diet concentration.

## Concluding remarks

Our work represents the first effort to understand the role of *naturally occurring variation* in the DR response and the mechanisms that underlie this variation. Studies such as this are critical to the future of health and aging research, as the most common age-related diseases are genetically and phenotypically heterogeneous, and, as a result, likely require treatments tailored to environment and genotype. As such, it is imperative to explore mechanisms of aging and disease not only in single, traditional laboratory strains, but also in the context of genetically heterogeneous animal models.

## Materials and methods

### Drosophila DR and lifespan

Methods for lifespan experiments were originally described in Wilson *et al.* [10]. Briefly, DGRP lines were obtained from the Bloomington *Drosophila* Stock Center. All flies were maintained on standard stock food (1.55% live yeast, 7.5% sugar, 8.5% cornmeal, 0.46% agar, 85% water). Flies were kept on a 12-hour light/dark cycle at 25˚C and approximately 65% humidity throughout the experiment. In preparation for lifespan measurements, approximately 15 female flies and 3 male flies were put in each bottle of stock food. Approximately 6 bottles were set up per strain. Five days after setting up stocks, adults were discarded, leaving

behind larvae that would be used for lifespan studies. Fourteen days after set-up when the experimental flies were estimated to be 2–3 days old, mated female flies were sorted into approximately 8 vials of either high yeast extract food (AL; 5% yeast extract, 5% sugar, 8.5% cornmeal, 0.46% agar, 85% water) or low yeast extract food (DR; same as previous, but with 0.5% yeast extract), with 25 flies per vial, targeting a total of 200 flies per strain*diet combination. Flies were transferred to fresh vials every other day. Recording of fly deaths commenced 8 days after flies started the new diet. All fly lifespan values presented here represent mean age-at-death measured in days from beginning the experimental diet.

Three flies per line and diet were frozen 5 days after beginning the experimental diet and shipped to the University of Washington in Seattle, WA for metabolomic profiling. We chose to sample at 5 days to try and capture early metabolomic indicators of DR response. Published studies from our group and others have shown metabolomic and other phenotypic differences in response to DR as early as 2–10 days after beginning the experimental diet [11, 43, 74].

## RNAi experiments

RNAi experiments were performed utilizing the Gal-UAS system [34]. Briefly, temporal knockdown of target genes was carried out using the ubiquitous drug-inducible GeneSwitch driver *Act5C*-GS-Gal4 and tissue specific knockdown was performed using the pan-neuronal driver *elav*-GS-Gal4, the fat body driver *S106*-GS-Gal4, or the gut-specific driver *5966*-GS-Gal4. Following development, flies were sorted onto AL or DR foods that contained 200uM RU486 to allow activation of the GeneSwitch system. Flies were maintained under these conditions throughout life.

## Metabolomics sample preparation

Fly samples were prepared following previously described procedure [75–77] and is detailed in our Supplementary Methods (S1 Text). Briefly, samples were thawed at room temperature, homogenized in 10:1 PBS:Water, methanol containing known concentrations of 6C13-glucose and 2C13-glutamate was added, samples were vortexed and stored at -20˚C for 20 min. Afterwards, samples were sonicated in an ice bath, centrifuged at 20,600 g and supernatant was recovered and dried. At the end, dried supernatant was reconstituted in a buffer containing known concentrations of 2C13-Tyrosine and 1C13-Lactate. A sample quality control (QC-S) was made by pooling small volumes of randomly chosen 30 prepared and reconstituted fly samples and this QC was used to monitor the data reproducibility.

## Liquid Chromatography–Mass Spectrometry (LC-MS)

Each sample was injected twice, 15 μL and 5 μL for analysis in negative and positive ionization modes, respectively. Both chromatographic separations were performed in HILIC mode on two parallel identical amide-based analytical columns. While one column was performing the separation, the other column was getting reconditioned and ready for the next injection. After the chromatographic separation, MS ionization and data acquisition were performed using an AB Sciex QTrap 5500 mass spectrometer (AB Sciex, Toronto, ON, Canada) equipped with an electrospray ionization (ESI) source and operating in multiple-reaction-monitoring (MRM) mode. We monitored 122 and 84 MRM transitions in negative and positive mode, respectively (206 MRM transitions total corresponding to 202 metabolites and 4 stable isotope-labeled internal standards). Additional details regarding LC-MS data acquisition, processing, and QC monitoring are included in the Supplementary Methods (S1 Text).

## Statistical analysis

All analyses described below were carried out using the open source software package R [78]. A false discovery rate of $\alpha = 0.01$ using the Benjamini-Hochberg-Yekutieli procedure [79] was used for all multiple comparisons unless otherwise stated. The metabolome data can be found in S1 Dataset, and the data used to make all the figures in this study can be found in S2 Dataset. A detailed description of these data are included in S1 Text.

**Calculating relative change in lifespan.** To monitor the effect of DR on lifespan, we use the relative change in lifespan (rLS) which was calculated by taking the residuals of a simple linear least squares regression of DR lifespan against AL lifespan (S1B Fig). The purpose of calculating rLS was to create a phenotype similar to the absolute different between DR and AL lifespan, but removes the potentially confounding effects of AL lifespans driving the lifespan response (Fig 1B and S1C and S1D Fig).

**Data normalization.** All metabolites were log-transformed to approximate a Gaussian distribution. Metabolites with >5% missing measurements were excluded from the analysis. Remaining missing data were imputed using the "impute" R package [80]. The final metabolomics dataset was comprised of 356 samples (178 fly lines x 2 diet conditions) with 105 individual metabolite features for each sample. Metabolite data were then mean-centered and unit-scaled within samples to normalize for variation in analyte amount (due to variation in fly weight/size) loaded onto the mass spectrometer. Flies were handled in different experimental batches and these experimental batches were found to have a statistically significant effect on many metabolite levels, but not on lifespan. To correct for the batch effect, we regressed out the effect of batch using a linear model. Depending on the type of analysis, we implemented 2 different types of batch correction: 1) correcting for batch within samples of the same diet, or 2) across all samples regardless of diet. The type of batch correction is indicated in the methods sections below.

**Multivariate and univariate analysis.** Principle component analysis (PCA) is a form of unsupervised multivariate analysis that partitions out independent variance components across a dataset of multiple, potentially correlated, variables. PCA was performed on all samples using the "prcomp" function in base R to observe how well the metabolome can separate samples by diet.

Differential analysis was performed on individual metabolites to determine which metabolites increase or decrease in response to diet. Batch correction was performed across all samples together regardless of diet for both differential analysis and PCA.

We used a multiple regression model to test for the effects of metabolite abundance on lifespan phenotypes. Genotype-wide mean body mass was used as a covariate in all models. Batch correction was performed on samples within diet for this analysis.

**Analysis of RNAi lifespan experiments.** We used a multivariate Cox Proportional Hazards regression to model survival as a function of diet (AL or DR), treatment (control or RNAi), and the interaction between diet and treatment. A significant interaction term suggested that the gene targeted by that RNAi treatment influences lifespan response to DR.

One of the assumptions of a Cox Proportional Hazards model is that the ratio of the hazards for individuals in different conditions remains constant over time, which can be evaluated by examining whether or not curves from different treatments cross over one another in a survival plot. While this assumption holds for most of our survival experiments, we acknowledge that for some of our survival experiments, that assumption does not hold (for example, Fig 5E), making some of the test results harder to interpret. However, we decided to apply the Cox model to all of our survival experiments regardless for consistency and interpretability.

**Network analysis and genome-wide association.**   For differential network analysis, we calculated Spearman's $\rho$ correlation matrices for all metabolites within each diet separately and used these to build unweighted differential metabolite networks. Batch correction was performed on samples within diet for this analysis.

To test the difference between the correlation structures under the two diets, we first calculated the difference in the two Spearman's $\rho$ correlation matrices (differential correlation matrix). We then tested whether the differential correlation of each pair of metabolites is significant by using parametric tests for Spearman's $\rho$ correlations, wherein we used the limiting normal distribution of the differential correlation for large sample sizes. More specifically, to improve normal approximations, we considered the scaled Spearman's $\rho$ for a single diet with $n$ samples, i.e., $\rho*sqrt((n\text{-}2)/(1\text{-}\rho^2))$. With independent samples, the difference in the scaled correlations between the two diets is approximately normal with mean = 0 and variance = 2. This asymptotic distribution was used to obtain $P$ values for each pair of differential correlations. The resulting $P$ values were adjusted for multiple testing.

Genotype variant calls for the DGRP are publicly available online (http://dgrp2.gnets.ncsu.edu/). Genome-wide association tests were performed in PLINK v.1.07 [81] using an additive linear model fitting phenotype (metabolite levels or lifespan response) as a function of variant, including *Wolbachia pipientis* infection status and major inversions as covariates. Variants with a minor allele frequency > 0.05 were included in the analysis. To correct for the observation that genes with a greater number of variants are expected by chance to have lower minimum $P$ values, we calculated a gene-specific "gene score". Gene scores for each gene were calculated by initially taking the minimum $P$ value from all the variants associated with a particular gene, including variants located within or 1,000 bp upstream or downstream of that gene. Following this, 1,000,000 permutations of this analysis for the top genes were implemented taking the minimum $P$ value per gene from of those permutations. Top genes were first identified if they had a preliminary minimum $P \leq 10^{-3}$ from an initial 10,000 permutations of all genes. Permutations were conducted by randomizing the genotype designation among all samples. The final gene score was then calculated by dividing the number of permutations where the observed minimum $P$ value was equal to or lower than the empirical minimum $P$ value by the total number of permutations. In effect, we are asking if the minimum $P$ value within a gene is even smaller than one would expect, given the number of variants found within that gene. A gene was then included in the network if it had a final gene score $\leq 10^{-4.5}$. All networks were visualized using Cytoscape [82].

## Supporting information

**S1 Fig. DGRP lifespan on AL and DR replotted from Wilson *et al*, 2020.** (A) Mean lifespan of 161 DGRP lines on AL (5% yeast extract) or DR (0.5% yeast extract) diets. Least squares linear regression of DR lifespan (B) and change in lifespan (DR–AL; C) and rLS (D) as a function of AL lifespan.
(PDF)

**S2 Fig. Diet restriction dramatically remodels the metabolome.** (A) PCA of all samples using metabolite profiles colored by diet. Ellipses are drawn with confidence level of 90%. (B) Volcano plot of significance of the difference between metabolite abundance with DR reveals that almost all metabolites are highly significantly changed with DR. Each point on the plot represents a single metabolite result from a pairwise Student's t-test.
(PDF)

**S3 Fig. Diet-dependent survival of GDH, GS1, and GS2 RNAi flies.** (A) The α-KG/gluta-mine pathway was manipulated by inhibiting transcript levels of *gs1/2* and *gdh*. (B-G) Survival of RNAi (+RU486) versus control (-RU486) flies of GS2, GS1, and GDH RNAi in whole-body (act5c-gal4-gs driver; B-D), GS2 RNAi in brain (elaV-gal4-gs driver; E), GS2 RNAi in fat body (S106-gal4-gs; F), and GS2 RNAi in gut (5966-gal4-gs; G) on AL and DR. Vertical lines indicate mean survival. Statistical model is a Cox Proportional Hazards model fitting survival as a function of diet, RNAi, and the interaction between diet and RNAi. Hazard ratios (HR) and P values are specific to the interaction term.
(PDF)

**S4 Fig. Multi-omic network for lifespan response for DR metabolites.** Gene-metabolite-phenotype network was constructed from linear modeling and GWAS results from DR metabolites that were correlated with lifespan response as measured by rLS. Gene nodes are colored in teal, metabolite nodes are colored in yellow. Gene node size is directly proportional to node degree, while metabolite node size is held constant. An edge exists between a metabolite and lifespan response if the metabolite was significantly correlated with rLS at a 1% FDR level of significance. Grey labels are metabolites correlated with rLS with no significant gene associations. An edge exists between a gene and metabolite and/or lifespan response if it the gene had a score of $\leq$1E-4.5.
(PDF)

**S5 Fig. Diet-dependent survival of RNAi fly strains.** Survival of inducible RNAi (+RU486) versus control (-RU486) flies of *selT* RNAi. Vertical lines indicate mean survival. Statistical model is a Cox Proportional Hazards model fitting survival as a function of diet, RNAi, and the interaction between diet and RNAi. Hazard ratios (HR) and P values are specific to the interaction term.
(PDF)

**S1 Table. All supplemental tables.**
(XLSX)

**S1 Dataset. Normalized metabolome data (within-diet and across-diet normalized data both included).**
(XLSX)

**S2 Dataset. Numerical data and summary statistics used to make main and supplementary figures.**
(XLSX)

**S1 Text. Supplementary Methods, summary statistics, and explanation of numerical data for figures.**
(PDF)

## Acknowledgments

This work was in part facilitated through the use of advanced computational, storage, and networking infrastructure provided by the Hyak supercomputer system and funded by the STF at the University of Washington.

## Author Contributions

**Conceptualization:** Rachel B. Brem, Pankaj Kapahi, Daniel Promislow.

**Data curation:** Danijel Djukovic.

**Formal analysis:** Kelly Jin.

**Funding acquisition:** Ali Shojaie, Pankaj Kapahi, Daniel Promislow.

**Investigation:** Kelly Jin, Kenneth A. Wilson, Jennifer N. Beck, Christopher S. Nelson, George W. Brownridge, III, Danijel Djukovic, Shiqing Yu.

**Methodology:** Kelly Jin, Kenneth A. Wilson, Jennifer N. Beck, Christopher S. Nelson, George W. Brownridge, III, Benjamin R. Harrison, Danijel Djukovic, Daniel Raftery, Rachel B. Brem, Shiqing Yu, Mathias Drton, Ali Shojaie, Pankaj Kapahi, Daniel Promislow.

**Project administration:** Pankaj Kapahi, Daniel Promislow.

**Software:** Shiqing Yu, Mathias Drton, Ali Shojaie.

**Supervision:** Pankaj Kapahi, Daniel Promislow.

**Validation:** Kenneth A. Wilson, Jennifer N. Beck, Christopher S. Nelson, George W. Brownridge, III, Danijel Djukovic, Shiqing Yu.

**Visualization:** Kelly Jin.

**Writing – original draft:** Kelly Jin.

**Writing – review & editing:** Kelly Jin, Kenneth A. Wilson, Jennifer N. Beck, Christopher S. Nelson, George W. Brownridge, III, Benjamin R. Harrison, Danijel Djukovic, Daniel Raftery, Rachel B. Brem, Shiqing Yu, Mathias Drton, Ali Shojaie, Pankaj Kapahi, Daniel Promislow.

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
