## [Decision Letter · Decision Letter 0]

26 Feb 2020

Dear Dr Jin,

Thank you very much for submitting your Research Article entitled 'Genetic and metabolomic architecture of variation in diet restriction-mediated lifespan extension in Drosophila' to PLOS Genetics. Your manuscript was fully evaluated at the editorial level and by independent peer reviewers. The reviewers appreciated the attention to an important topic but identified some aspects of the manuscript that should be improved.

We therefore ask you to modify the manuscript according to the review recommendations before we can consider your manuscript for acceptance. Your revisions should address the specific points made by each reviewer.

[LINK]

Yours sincerely,

Coleen T. Murphy

Associate Editor

PLOS Genetics

Gregory P. Copenhaver

Editor-in-Chief

PLOS Genetics

The reviewers were overall pleased with the paper, and had minor suggestions to improve some aspects. Please address these concerns before resubmitting the manuscript.

Reviewer's Responses to Questions

**Comments to the Authors:**

Reviewer #1: In the present study, Jin. et al., aim to understand how the response to life-extending interventions varies between genotypes. Previously, dietary restriction (DR) has been shown to extend lifespan in most model organisms. However, studies in yeast, flies and mice demonstrated extensive within-organism variation between varying genetic backgrounds. The mechanism for this difference remains understudy, a key question if basic biology of aging is to be extended to generate usable human therapeutics. Therefore, the authors applied a systems biology approach to build candidate gene-to-metabolite-to-phenotype pathways that may underly the differential response to DR in different strains of inbred flies. The authors identify specific metabolic hubs that explain DR response variation, as well as gene-metabolite pairs strongly associated with DR lifespan extension. Additionally, the authors validated two of their gene hits using RNAi lifespans.

Overall, this study is very timely and provides insight into the mechanisms underlying natural variation in DR response, which will likely open doors for further mechanistic studies. These data will be of interest to both the aging and systems biology fields, along with more broadly those interested in precision medicine type approaches to gene/environment interactions.

Main points:

o In the data set used around 30% of the population DR didn’t extend lifespan, around half of these had their lifespan shorten with DR. In addition to using mean lifespan and relative lifespan change as a measurement to capture the overall effect, might it give more detection power to extend the analysis to bins of subgroups with differential effect directionality (i.e. responders, non-responders, responded negatively) and look for significant differences between those binned groups?)

o The authors did the metabolomics and gene-level GWAS data collection on flies at Day 5 after diet started. I would recommend including a discussion about the timing of this measurement; when is the onset of metabolic changes associated with DR thought to be?

o The authors identified genes (for example, SNP in CCHaR) and metabolites (for example, alpha-ketoglutarate) strongly associated with lifespan, and validated them using RNAi. One of the strengths of the paper is the identification of gene-metabolite interactions that are associated with a phenotype. However, the addition of a targeted metabolomics measurement in the validation experiment could strengthen the conclusions – ie looking at the impact of DR +/- knock down of CCHaR.

Minor points

o In lifespans (Figures 5, S3, S5), it would be beneficial to include the mean lifespan line, and report the effect measurement. The effect seems to be more of an increase in median lifespan versus maximum lifespan; revise when reporting the results and drawing conclusions.

o Throughout the graphics, the authors should highlight the metabolites and genes that are discussed, to make it clearer to the reader where the hits come from.

o In page 13, 5th sentence the connector ‘and’ is missing from the phrase ‘peripheral organs the brain’.

Reviewer #2: This is a very strong, highly integrative submission dealing with one of the most central problems in the biology of aging: by what mechanism does dietary restriction extend lifespan. The results are uniquely informative. The synthetic, systems biology approach is ground-breaking. Interested readers will span those from the medical sciences, neurobiology, longevity research, nutritional research, computational biology, genetics and systems biology. I find no essential faults with the design or analysis. I think the conclusions are well supported and within the scope of the data. I especially like the validation work where candidate hubs are tested by RNAi manipulative experiments. This is the way to move from GWAS to inference on mechanism. Naturally, I am intrigued by the various hits and proposed regulatory pathways for control of variation in the dietary restriction response. These leads will stimulate substantial research.

I have only a few editorial or aside remarks. The hardest problem will involve the ability to generalize the outcomes given the way diet was manipulated. This issue needs community level work.

Please clarify the data usage and relationship with the Wilson manuscript. I cannot tell if the survival data and sampled flies of this PLoS submission are de novo material unique to the PLoS submission, or if these data and flies were actually one and the same as those presented in Wilson. Please clarify as well, that Wilson was uploaded to BioRxiv. I would not call this “originally published” as stated on page 6. Or, perhaps the Wilson work has actually been peer reviewed and ‘published’ by now. Using the flies and data in two papers is fine, but simply would require clarification. Finally, it seems that the Wilson project describes several candidate variant genes to affect diet dependent longevity, yet the present work finds perhaps only one (CG6231).

On page 9, the text (first paragraph) mentions “significance scores for each gene-metabolite pair”. Readers are referred to the methods for details. I recommend adding a sentence to define these scores here; this will aide in understanding the paragraph without putting the reader in a bind to figure out details on their own in the M&M.

The paper acknowledges ‘Limitations of study’. I like this frankness. An important issue, as noted, will be the method used to make diet based on yeast extract. Given this method, the results and conclusions seem very robust and real. The problem will be that most (all) other labs use processed whole yeast in adult diets, not yeast extract, and the authors know about this controversy. It is a positive move to acknowledge the issue, but this does not fix the problem or establish the generality of their mechanistic conclusions. I am not sure what to recommend here but I can see the paper will always have an asterisk until its results are verified using more conventional diets, perhaps limiting these tests to the candidate gene RNAi studies.

Page 16. Please elaborate and clarify the logic for how residuals are used make the rLS. I don’t understand how residuals ‘remove potentially confounding effects of extreme AL …” Residuals are calculated for all points, and adjusts each observation relative to its expected value based on linear regression. Why is this approach used instead of, perhaps, the ratio of DR to AL median longevity?

**Have all data underlying the figures and results presented in the manuscript been provided?**

Reviewer #1: Yes

Reviewer #2: Yes

PLOS authors have the option to publish the peer review history of their article (what does this mean?). If published, this will include your full peer review and any attached files.

Reviewer #1: No

Reviewer #2: No

---

## [Decision Letter · Decision Letter 1]

6 May 2020

Dear Dr Jin,

We are pleased to inform you that your manuscript entitled "Genetic and metabolomic architecture of variation in diet restriction-mediated lifespan extension in Drosophila" has been editorially accepted for publication in PLOS Genetics. Congratulations!

Yours sincerely,

Coleen T. Murphy

Associate Editor

PLOS Genetics

Gregory P. Copenhaver

Editor-in-Chief

PLOS Genetics

Comments from the reviewers (if applicable):

Reviewer's Responses to Questions

**Comments to the Authors:**

Reviewer #1: The authors have addresses all of our concerns

Reviewer #2: A good job made even better.

**Have all data underlying the figures and results presented in the manuscript been provided?**

Reviewer #1: Yes

Reviewer #2: Yes

PLOS authors have the option to publish the peer review history of their article (what does this mean?). If published, this will include your full peer review and any attached files.

Reviewer #1: No

Reviewer #2: No

**Data Deposition**

http://datadryad.org/submit?journalID=pgenetics&manu=PGENETICS-D-20-00039R1

**Press Queries**

---

## [Editor Report · Acceptance letter]

21 May 2020

PGENETICS-D-20-00039R1 

Genetic and metabolomic architecture of variation in diet restriction-mediated lifespan extension in *Drosophila*

Dear Dr Jin, 

We are pleased to inform you that your manuscript entitled "Genetic and metabolomic architecture of variation in diet restriction-mediated lifespan extension in *Drosophila*" has been formally accepted for publication in PLOS Genetics! Your manuscript is now with our production department and you will be notified of the publication date in due course.

With kind regards,

Matt Lyles

PLOS Genetics

On behalf of:
